# Prenatal Bisphenol a Exposure, DNA Methylation, and Low Birth Weight: A Pilot Study in Taiwan

**DOI:** 10.3390/ijerph18116144

**Published:** 2021-06-07

**Authors:** Yu-Fang Huang, Chia-Huang Chang, Pei-Jung Chen, I-Hsuan Lin, Yen-An Tsai, Chian-Feng Chen, Yu-Chao Wang, Wei-Yun Huang, Ming-Song Tsai, Mei-Lien Chen

**Affiliations:** 1Department of Safety, Health and Environmental Engineering, National United University, Miaoli 360, Taiwan; yfh@nuu.edu.tw; 2Center for Chemical Hazards and Environmental Health Risk Research, National United University, Miaoli 360, Taiwan; 3Institute of Food Safety and Health Risk Assessment, National Yang Ming Chiao Tung University, Taipei 112, Taiwan; 4School of Public Health, Taipei Medical University, Taipei 110, Taiwan; koko826@gmail.com; 5Institute of Environmental and Occupational Health Sciences, School of Medicine, National Yang Ming Chiao Tung University, Taipei 112, Taiwan; polly06011991@yahoo.com.tw (P.-J.C.); b507092003@tmu.edu.tw (Y.-A.T.); 6VYM Genome Research Center, National Yang Ming Chiao Tung University, Taipei 112, Taiwan; ycl6.gel@gmail.com (I.-H.L.); cfchen@ym.edu.tw (C.-F.C.); 7Institute of Biomedical Informatics, National Yang Ming Chiao Tung University, Taipei 112, Taiwan; yuchao@ym.edu.tw; 8Immuno Genomics Co., Ltd., Taipei 112, Taiwan; richard0609@gmail.com; 9Department of Biological Science and Technology, National Yang Ming Chiao Tung University, Taipei 112, Taiwan; 10Institute of Bioinformatics and Systems Biology, National Yang Ming Chiao Tung University, Taipei 112, Taiwan; 11Department of Obstetrics and Gynecology, Cathay General Hospital, Taipei 110, Taiwan; 12School of Medicine, Fu Jen Catholic University, New Taipei 242, Taiwan

**Keywords:** bisphenol A, DNA methylation, Illumina HumanMethylation 450 BeadChip, birth outcomes

## Abstract

Prenatal exposure to bisphenol A (BPA) may increase the risk of abnormal birth outcomes, and DNA methylation might mediate these adverse effects. This study aimed to investigate the effects of maternal BPA exposure on maternal and fetal DNA methylation levels and explore whether epigenetic changes are related to the associations between BPA and low birth weight. We collected urine and blood samples originating from 162 mother-infant pairs in a Taiwanese cohort study. We measured DNA methylation using the Illumina Infinium HumanMethylation 450 BeadChip in 34 maternal blood samples with high and low BPA levels based on the 75th percentile level (9.5 μg/g creatinine). Eighty-seven CpGs with the most differentially methylated probes possibly interacting with BPA exposure or birth weight were selected using two multiple regression models. Ingenuity pathway analysis (IPA) was utilized to narrow down 18 candidate CpGs related to disease categories, including developmental disorders, skeletal and muscular disorders, skeletal and muscular system development, metabolic diseases, and lipid metabolism. We then validated these genes by pyrosequencing, and 8 CpGs met the primer design score requirements in 82 cord blood samples. The associations among low birth weight, BPA exposure, and DNA methylation were analyzed. Exposure to BPA was associated with low birth weight. Analysis of the epigenome-wide findings did not show significant associations between BPA and DNA methylation in cord blood of the 8 CpGs. However, the adjusted odds ratio for the dehydrogenase/reductase member 9 (DHRS9) gene, at the 2nd CG site, in the hypermethylated group was significantly associated with low birth weight. These results support a role of BPA, and possibly DHRS9 methylation, in fetal growth. However, additional studies with larger sample sizes are warranted.

## 1. Introduction

Accumulating evidence from animal and epidemiological studies has implied that the in utero and early-life environments play critical roles in a variety of adverse health outcomes [1]. This observation derives from the developmental origins of health and disease (DOHaD) hypothesis (i.e., Barker’s hypothesis), stating that the intrauterine environment can “program” the fetus to disease predisposition later in life by making subtle changes in organ structure or function [2]. There is increasing concern that several environmental pollutants and lifestyles are risk factors for adverse prenatal development, pregnancy outcomes, and offspring health [3]. Bisphenol A (BPA), an endocrine-disrupting chemical (EDC), has received substantial attention because of its widespread use in consumer products [4] and induction of increased risks for low birth weight, adverse birth outcomes, metabolic disorders, and cancers in rodents and humans [5,6,7,8,9]. BPA was detectable in urine samples from pregnant women and fetal cord blood samples in a Taiwanese population, indicating that exposure to BPA is common in Taiwan [8,10].

Growing evidence suggests that epigenetic mechanisms play a crucial role in modulating exposure-disease relationships [11]. DNA methylation, a naturally occurring modification that involves the addition of a methyl group to the 5′ position of the cytosine ring, is a commonly studied epigenetic marker. Methylation, occurring predominantly at cytosine-guanine dinucleotide (CpG) sites, regulates gene expression and maintains genomic stability in the human genome [12]. Animal studies have demonstrated that DNA methylation changes in the offspring can occur in response to BPA exposure [6]. An epigenetic epidemiological study showed that prenatal BPA exposure causes hypomethylation in the obesity-associated mesoderm-specific-transcript (MEST) gene promoter in cord blood as well as increased expression of MEST, which contributes to increases in the body mass index (BMI) Z-score in children until the age of 6 years [13]. In another epigenome-wide association study on children, being in the high BPA exposure group, compared to the low BPA exposure group, categorized by the 80th percentile of maternal BPA levels (2.7 μg/g creatinine), was associated with an increase in DNA methylation of insulin-like growth factor 2 receptor (IGF2R). Notably, the increased methylation of IGF2R at age 2 was associated with BMI during age 2–8 years only in girls [14]. Although previous human studies have reported associations between BPA exposure and DNA methylation alteration in female blood [15], clinical fetal tissues samples [16], and cord blood [17], data on the role of maternal BPA exposure in relation to maternal/fetal DNA methylation in the blood and epigenetic changes linked to newborn birth outcomes are limited. Thus, we were motivated to explore the epigenetic effects of BPA exposure on DNA methylation and elucidate the effects of methylation on low birth weight. We aimed to examine whether prenatal BPA exposure and DNA methylation related to low birth weight.

To achieve this goal, we first assessed urine samples originating from 162 mother-infant pairs in a Taiwanese cohort study, wherein 34 pregnant women who had high (*n* = 8) and low BPA (*n* = 26) levels in all three trimesters were selected, based on urinary BPA levels stratified by the 75th percentile (9.5 μg/g creatinine) [7]. We first measured 34 maternal blood DNA methylation levels using the Illumina Infinium HumanMethylation 450 BeadChip, which allowed measurement of the methylation levels at 485,577 CpG sites. In step one, we selected the most differentially methylated candidate CpGs using two multiple regression models. In step two, we utilized ingenuity pathway analysis (IPA) to select 18 CpG sites related to the disease categories “developmental disorder, skeletal and muscular disorders, skeletal and muscular system development, metabolic disease and lipid metabolism” and then validated these sites by pyrosequencing 82 cord blood samples. In step three, a causal-mediation analysis was utilized to evaluate whether the risk of low birth weight related to BPA exposure is explained by the cord DNA methylation of 8 CpG sites (Figure 1).

## 2. Material and Methods

### 2.1. Study Participants and Sample Collection

A total of 162 mother-fetus pairs were recruited from an obstetrics clinic at Cathay General Hospital (CGH) in Taiwan who reported for Down’s syndrome screening in 2010. The study protocol was approved by the Institutional Review Board of CGH in Taipei, Taiwan, and details regarding the study population have been published elsewhere [7]. The eligibility criteria included that the participants were aged 18–45 years, were less than 13 weeks pregnant with detection of the fetal heartbeat at the first prenatal visit, and were planning to deliver at CGH. We excluded fetuses with structural abnormalities or chromosomal defects. Women provided three spot urine samples and three blood samples at approximately 11 (range 10–13) and 26 (range 24–28) weeks gestation and at admission for delivery, and they completed a structured questionnaire in the first and third trimesters. We followed the participants until delivery, collected cord blood samples, and recorded neonatal birth weight (g). From this population, 34 maternal blood samples in the three trimesters were selected, based on urinary BPA levels stratified by the 75th percentile as having high BPA (mean of 34.53 (range 12.76–114.95) μg/g creatinine, *n* = 8) and low BPA (mean of 0.71 (range 0.01–8.37) μg/g creatinine, *n* = 26) in all three trimesters, to explore the association among prenatal BPA exposure, DNA methylation, and low birth weight. 

### 2.2. Measurement of BPA and Creatinine

Urinary creatinine measurements were obtained for all 162 women. Due to limited urine volume, 144, 137, and 121 urine samples were analyzed for BPA in the first, second, and third trimesters, respectively. Sample pretreatment and quantification of BPA followed previous procedures [10]. In brief, urine samples (two milliliters) were spiked with 50 μL of ^13^C_12_-BPA (2000 ng/mL), acidified to pH 5.5, deconjugated by *β*-glucuronidase (Sigma-Aldrich, Saint Louis, MO, USA), incubated at 37 °C for 15 h, and then acidified to pH 3 with 1 M hydrochloric acid. Next, the samples were extracted with a PH solid-phase extraction cartridge and were eluted with 1 mL methanol. BPA was analyzed using ultra-performance liquid chromatography coupled with time-of-flight mass spectrometry (Waters, Milford, MA, USA). Separation was achieved using a BEH C18 column (1.7 μm, 2.1 × 10 mm, Waters, Milford, MA, USA) with a flow rate of 0.35 mL/min. Chromatography was performed using binary mixtures of 0.1% ammonia in water (mobile phase A) and 0.1% ammonia in methanol. The elution gradient conditions were as follows: 80% A for 0.5 min, 80–1% A for 0.5 min, 1% A held for 1 min, 1–80% A for 1 min, and re-equilibration at 80% A for 3 min. The TOF/MS was operated by monitoring ion mass transitions as follows: m/z 227.12→133.07 for BPA and m/z 239.18→139.09 for ^13^C_12_-BPA. The average recovery for BPA was 93–100% at the 20–100 ng/mL level. The limit of detection (LOD) for BPA in urine was 0.16 ng/mL. Creatinine was analyzed based on modifications of the Jaffe reaction made by Hinegard and Tiderstrom [18]. Briefly, 0.1 mL of each urine sample was added to 3 mL 3.3 mM picric acid and mixed with 0.17 M sodium hydroxide and 26 mM sodium tetraborate. This mixture was then incubated at 37 °C for 15 min and measured with a spectrophotometer at a wavelength of 510 nm.

### 2.3. Quantification of DNA Methylation and Normalization

The maternal buffy coat was processed and immediately stored at −80 °C. DNA was extracted using the Genomic DNA Mini kit buffy coat protocol (Geneaid), and the DNA quality was assessed using spectrophotometry and gel electrophoresis. Genome-wide DNA methylation was assessed as follows: Isolated DNA (1000 ng) was bisulfite-converted using the EZ Methylation kit (Zymo Research, Irvine, CA, USA). Following bisulfite conversion, DNA was hybridized to the Infinium HumanMethylation 450 (450K) BeadChip (Illumina Inc. San Diego, CA, USA) and scanned according to the manufacturer’s protocol. Raw data files in the IDAT format were processed with the Chip Analysis Methylation Pipeline (ChAMP) Bioconductor package (v1.8.0) in the R environment. Probes meeting any of the following conditions were removed: (1) internal controls, (2) a detected *p*-value > 0.01, (3) < 3 beads in at least 5% of the samples per probe, (4) alignment to multiple locations, (5) demonstration of SNPs, or (6) an X chromosome locatqion. A total of 433,523 probes passed the filtering process, and intra-array normalization was performed with the BMIQ (Beta MIxture Quantile dilation) method. Finally, the beta values were batch-corrected using the ComBat method. 

### 2.4. Data Analysis

#### 2.4.1. Screen and Selection of Candidate CpGs

##### Screen Candidate CpGs by Multivariate Linear Regression

To identify the differentially methylated regions, we investigated the relationships among prenatal BPA exposure, maternal DNA methylation, and birth weight using the Matlab Limma package, which fits two multivariate linear regression models. We considered covariates including maternal age, pre-pregnancy body mass index (BMI), gestational age (GA), weight gain, infant sex, and pregnancy complications and diseases. First, using the maternal DNA methylation (DNA-M) of each CpG locus as the dependent variable, we utilized prenatal BPA exposure as the explanatory variable while also including the covariates mentioned above (DNA-M = a0 + a1 BPA + a2 maternal age + a3 pre-pregnancy BMI + a4 GA + a5 weight gain + a6 infant sex + a7 pregnancy complications and diseases). Second, using newborn birth weight (BW) as the dependent variable, we utilized the maternal DNA methylation of CpG sites as the explanatory variable and the covariates mentioned above (BW = a0 + a1 DNA-M+ a2 maternal age + a3 pre-pregnancy BMI + a4 GA + a5 weight gain + a6 infant sex + a7 pregnancy complications and diseases). Finally, 87 candidate CpG*s* were selected based on the two models, with β estimates at *p*-values < 0.05. For multiple comparisons, the *p* value was adjusted by the false discovery rate (FDR), which was a FDR-corrected *p* value, according to the Benjamini and Hochberg (1995) method [19].

##### Selection of Candidate Genes by Ingenuity Pathway Analysis (IPA) 

For biological insights, the genes of interest the categories of “developmental disorder, skeletal and muscular disorders, skeletal and muscular system development, metabolic disease and lipid metabolism” were analyzed for over-represented biological functions, diseases, and pathways in IPA (Qiagen, Redwood, CA, USA). 

### 2.5. Analysis of Methylation by Pyrosequencing and Validation in Cord Blood Samples 

Genomic DNA from cord blood samples of all subjects was extracted using the Genomic DNA Mini kit buffy coat protocol (Geneaid), and the DNA quality was assessed using spectrophotometry and gel electrophoresis. Based on IPA, 18 CpGs (17 genes) were selected, of which 8 CpGs (7 genes) were validated to have met the primer design score requirements using bisulfite treatment and quantitative PCR (q-PCR). Due to qualified DNA quality, 82 cord blood samples were bisulfite-modified using the EZ DNA Methylation kit (Zymo Research, Irvine, CA, USA). The primers used for bisulfite sequencing were designed with MethPrimer (Appendix A). The associations among BPA exposure, DNA methylation of candidate CpGs in cord blood, and low birth weight were further explored using causal-mediation analysis. Statistical analysis was performed using R software version 3.2.2 (The R Foundation for Statistical Computing, Vienna, Austria). Analyses were considered statistically significant at *p* < 0.05. Causal-mediation analysis was conducted to evaluate whether low birth weight related to BPA exposure is explained by the cord blood DNA methylation of 8 CpG sites. Three models (mediator, outcome, and mediation analysis) were included in the analyses. First, in the outcome model (model 1), using newborn birth weight as the dependent variable, we utilized BPA exposure as the explanatory variable as well as covariates, including the DNA methylation of CpG sites in cord blood, maternal age, pre-pregnancy BMI, GA, infant sex, and parity. Second, in the mediator model (model 2), using the cord blood DNA methylation of each CpG locus as the dependent variable, we utilized prenatal BPA exposure as the explanatory variable and included the same covariates as those used in model 1 with the exception of DNA methylation of CpG sites in cord blood. Third, in the mediation analysis model (model 3), using low birth weight as the dependent variable, we utilized BPA exposure as the explanatory variable and DNA methylation of CpG sites in cord blood as the mediator. This model incorporated four regression coefficients (indirect/mediated, direct, and total effects as well as the mediation proportions). The corresponding 95% confidence intervals (CIs) were calculated using 1000 bootstrap sampling. We applied logistical linear regression models to investigate the relationships between the cord blood DNA methylation levels at 8 CpG sites and low infant birth weight after adjusting for the covariates.

## 3. Results

The demographic characteristics of the study population with the geometric mean BPA levels are provided in Table 1. There were no significant differences between those with Illumina DNA methylation levels and those without. The subgroup of 34 pregnant women was selected from all 162 women, based on urinary BPA levels stratified by the 75th percentile (high mean (range) levels with 34.53 (12.8–114.9) μg/g creatinine; *n* = 8 and low mean (range) levels with 0.71 (0.01–8.4) μg/g creatinine; *n* = 26). To identify the differential methylated CpGs, two multivariate linear regression models were applied to find epigenome-wide significant associations between BPA exposure and maternal DNA methylation, and maternal DNA methylation and birth weight. Volcano plots showed *p* values versus the magnitude of effect (partial regression coefficient) on maternal DNA methylation with prenatal exposure to BPA and on birth weight with maternal DNA methylation (Appendix A). Eighty-seven candidate CpG sites were identified by the simultaneously associations corrected *p* value < 0.05 with two models (Appendix A).

Next, to identify biological processes associated with the identified BPA exposure-related methylation sites, we performed core analysis using IPA. As shown in Table 2, 18 CpG sites related to developmental disorders, skeletal and muscular disorders, skeletal and muscular system development, metabolic disease, and lipid metabolism were selected. Of these, 8 CpG sites (cg01502353: DST, cg05524038: CSF1R, cg07349217: TG, cg19427642: KCNMA1, cg19768311: DST, cg23244463: KCNB2, cg27420224: HNF4A, cg27640254: DHRS9) met the primer design score requirements and were selected and subjected to further testing in 82 cord blood samples. The DNA methylation percentages of 8 CpG sites in cord blood are presented in Table 3. The mean methylated CpGs percentages were 79–97% for 8 CpGs.

To test whether DNA methylation is a mediator between in utero BPA exposure during three trimesters and low birth weight, we conducted causal-mediation analysis. After controlling for covariates, each natural-log increase in urinary BPA level (μg/g creatinine) in the second trimester was associated with a 0.02 g decrease in low birth weight (*p* < 0.05). Nonsignificant associations were observed between BPA exposure and cord DNA methylation at the 8 CpG loci. Mediation analysis revealed that cord DNA methylation at the 8 CpG islands we selected may not mediate the effects of BPA exposure on low birth weight (Appendix A). The risk of low birth weight was evaluated by stratifying the cord DNA methylation levels at 8 CpG loci into high and low groups based on the 50th percentile. Table 4 shows that after controlling for covariates, the adjusted odds ratio for the DHRS9 gene, at the 2nd CpG site in the hypermethylated group, was significantly associated with low birth weight.

However, we did not find associations between BPA and DNA methylation in cord blood at the 8 CpG sites and failed to establish an epigenetic link between BPA and low birth weight. A considerable degree of normal within- and between-individual variability is observed in the assessment of maternal and cord DNA methylation using the Illumina assay that is not well understood. In the present study, we observed nonsignificant associations between maternal and cord DNA methylation, except for a significant correlation between maternal and cord blood DNA methylation at 1 CpG site, cg01502353 (DST gene; *n* = 12, *r*sp = 0.73, *p* = 0.01; data not shown). Further studies with cord blood DNA methylation using the Illumina assay are needed to investigate the potential epigenome responses to BPA exposure.

## 4. Discussion

To our knowledge, this is the first epidemiological study linking in utero BPA exposure and low birth weight with an epigenetic marker (i.e., DNA methylation) as the underling mechanism. In this study, we found significant associations between BPA in the second trimester, which represents a critical window of susceptibility for fetal development [7], and low birth weight, and between DHRS9 methylation in cord blood and low birth weight. These results suggest that prenatal BPA exposure and cord blood DNA methylation of DHRS9 may play roles in fetal growth. 

The geometric mean (GM) of the in utero BPA levels in our population was slightly lower than those reported in pregnant women in the USA [20], the Netherlands [21], Canada [22], Spain [23], Australia [24], and France [25]. The difference in BPA levels in individuals from Taiwan and other countries might be caused by the lifestyles in each country. One study reported the order of BPA levels in tissue distribution (e.g., maternal serum and urine, neonatal urine and cord serum) are urine in mother and neonates > cord serum > maternal serum [26]. In addition, BPA in cord serum was significantly associated with maternal serum and urine. Amniotic fluid is mainly derived from fetal urine and the mass exchange with the umbilical cord during the 3^rd^ trimester. The level of BPA in amniotic fluid is lower than that in maternal urine, suggesting that the metabolism of BPA was low in the fetus [27,28]. In this study, the increasing trend of BPA throughout the three trimesters may be attributed to the changes in dietary behavior (e.g., the intake of meat, vegetables, and fruit) and lifestyle (e.g., frequency of using plastic products and detergent). Epidemiological studies assessing BPA in relation to birth weight have shown inconsistent results. Our observed significant association between the BPA level in the second trimester and low birth weight is similar to a previous study reporting decreases in birth weight [8,9,29] and other studies showing an increase in the ponderal index [30] or null associations [31,32,33]. The discrepancies between the results presented herein and previous results may be due to differences in the timing of sample collection during pregnancy, the use of a single spot sample, or the use of urine or blood samples for exposure assessment. 

Studies have reported three major epigenetic mechanisms of EDC exposure-induced fetal reprogramming, including genomic and non-genomic actions of ligand-activated transcription factors (TFs) and altered cellular environments [34]. Among them, investigations of ligand-activated TFs using non-genomic (i.e., signal transduction) mechanisms that potentially alter the epigenetic machinery involved in DNA methylation or histone post-translational modifications are ongoing. DNA methylation is critical during embryonic development, genomic imprinting, and X chromosome inactivation [35], and is established during embryogenesis and early fetal growth [36]. This study examined DNA methylation in cord blood samples because of the epigenome status of fetus [37] and conducted a causal-mediation analysis, revealing that the DNA methylation of 8 CpG sites may not mediate the linkage between BPA exposure and low birth weight. We believe that this result may be due to the absence of associations found between BPA and DNA methylation in cord blood, which is consistent with a study reporting no association between maternal BPA and the DNA methylation of *IGF2* in placental tissue [38]. In addition, other genes are likely affected by prenatal BPA exposure, and further investigations are warranted. 

The DNA methylation levels of 8 CpG sites in cord blood are consistent with those reported in the literature [39,40], and the impact of prenatal BPA exposure on DNA methylation is of great concern. In contrast, previous studies reported that BPA was capable of inducing DNA methylation in both animals [6,41] and humans [15,16,17,40,42]. Hanna et al., (2012) reported that BPA exposure was associated with DNA methylation changes in whole blood using Illumina GoldenGate Methylation Cancer Panel I bead array assays on 1505 CpG sites, in women undergoing ovarian stimulation for in vitro fertilization [15]. Nahar et al. (2015) revealed that BPA levels were positively associated with hypermethylation in 2nd trimester placental tissue only using the long interspersed nuclear element-1 (LINE-1) assay [16]. Miura et al., (2019) reported that BPA may induce DNA methylation status at birth in a sex-specific manner [17]. We speculated that this discrepancy was due to differences in the cell- or tissue-specific nature of the DNA methylation profiles, the DNA methylation methods, or the limited sample sizes. Furthermore, 8 of 18 CpG sites were selected because they met the primer design score requirements, which may have also contributed to the discrepant results. 

We chose to focus on seven growth-related genes (DST, CSF1R, TG, KCNMA1, KCNB2, HNF4A, DHRS9) in cord blood based on IPA analyses, and we could not exclude the DNA methylation of additional growth-related genes. Previous studies have shown that DNA methylation of HNF4A and MC4R (melanocortin 4 receptor) was associated with triglyceride levels in cord blood of preterm infants [40] and metabolic profiles in the blood of children aged 7–9 years [39]. Prior research has demonstrated associations between fetal growth and the DNA methylation of two growth-related genes (*IGF2* and aryl-hydrocarbon receptor repressor) [43]. Interestingly, considering candidate epigenetic markers for neonatal endpoints, we have provided evidence of a significant association between the hypermethylated DHRS9 gene and low birth weight. Dehydrogenase/reductase (SDR family) member 9 (DHRS9) is the third member of the short-chain dehydrogenases/reductases (SDR) 9C family and a protein coding gene (Genecards). This protein activates lipid metabolism, demonstrates oxidoreductase activity toward hydroxysteroids, and may function as a transcriptional repressor in the nucleus. Alternative splicing results in multiple transcript variants. We did not, however, conduct a gene expression analysis of DHRS9, and we thus have no information on whether DHRS9 is highly methylated and cannot establish a link with low birth weight. 

Four animal studies have reported that BPA exposure can provoke intrauterine growth restriction [44], cause preeclampsia-like features in pregnant mice involved in the reprogramming of DNA methylation of WNT2 (Wnt family member-2) [45], and cause toxicity in spermatocytes, and DNA methylation may exert an important role in BPA-triggered male reproductive toxicity [46] and alter the methylation levels of differentially methylated regions (DMRs) including the *Snrpn* imprinting control region and IGF2R DMR1 [47]. In addition, three human and epidemiological studies have shown that BPA can cause abnormal development of the central and peripheral nervous system in the fetus [48], and BPA exposure on children with asthma might operate through the alteration of *MAPK1* methylation [42]. In another epigenome-wide association study on children, being in the high BPA exposure group, compared to the low BPA exposure group, was associated with an increase in DNA methylation of IGF2R [14]. Although previous experimental and human epidemiologic studies reported the effects of BPA exposure on fetal development and DNA methylation, there were different approaches of methodology or the interested endpoint of health from our present study. Data on the role of maternal BPA exposure in relation to maternal/fetal DNA methylation in the blood and epigenetic changes linked to newborn birth outcomes are very limited. This study has several strengths. First, we used multiple urine sampling points during pregnancy to explore the exposure to BPA and a critical window of susceptibility for fetal development. Second, we applied a novel approach: using an Illumina assay on multiple blood samples to identify candidate CpG methylation sites, followed by IPA and confirmation of the candidate CpG methylation sites by pyrosequencing. This step-wise approach is cost-effective and can help determine the most promising CpG sites responsible for the association between BPA exposure and low birth weight. We also acknowledge some limitations of this study. First, the sample sizes for the analyses were relatively small, limiting our power to explore associations between exposure-health or exposure-DNA-methylation relationships. Second, we screened 17 candidate CpG sites, finally selecting 8 CpG sites for validation. Thus, we cannot exclude other CpG sites linking BPA exposure and low birth weight, possibly impacting our findings. Third, because we were unable to obtain RNA, RNA expression patterns were not examined to explore the influence of DNA methylation on gene expression. Fourth, a technical validation of DNA methylation patterns in mothers and fetuses was not performed. The interpretation of DNA methylation data from whole blood should be cautious and particularly across different stages throughout pregnancy. Even though we did not validate the 450 K assay in maternal samples, we used a multivariate linear regression model to identify those differentially methylated CpG sites which were significantly associated with BPA exposure and birth outcomes. Thereby, IPA was employed to select candidate genes related to system developmental to examine the association among DNA methylation levels in cord blood, prenatal exposure to BPA, and low birth weight. Fifth, cell counts were unavailable and so there was no information on whether specific white blood cells were highly methylated and could establish the linkage with birth weight. The interpretation of DNA methylation profiles from whole blood should be cautious, because the differences might be caused by varying proportions of cell types. The fluorescence-activated cell-sorting approach may be needed to isolate lymphocytes, monocytes, granulocytes, and neutrophils. Since different types of white blood cells might be responsible for the DNA methylation changes [49], the confounding effect of cell proportion needs to be adjusted. It is no doubt that cord blood differs from adult peripheral blood in the aspects of cell type compositions and distinct cell type specific differences in DNA methylation [49]. The multivariate linear regression model applied to the methylation changes analysis in cord blood is more appropriate than in maternal blood. Unfortunately, the sample number of cord blood is too small (*n* = 12). There is distinct cell-type-specific DNA methylation in cord blood, reflecting the hematopoietic lineage; however, there were few inter-individual cell type specific differences in DNA methylation [49]. Meanwhile, the selected candidate genes by IPA are biologically functionally related to system development. Therefore, the candidate genes and methylation levels of CpG sites in cord blood were studied to examine their correlation among BPA exposure and birth weight. Since the study subjects of newborns were healthy, there was reasonably little inter-individual variation in the cell type compositions [49]. Sixth, we were unsure of which pregnancy trimesters in pregnant women or fetuses are targeted by DNA methylation. 

## 5. Conclusions

In conclusion, our results add to the evidence that BPA exposure in the second trimester or the hypermethylated DHRS9 gene in cord blood is associated with low newborn birth weights. Future studies with larger sample sizes and different types of cell or tissue samples are required to unveil the potential epigenetic mechanisms by which BPA affects DNA methylation and their effects on birth outcomes.

## Figures and Tables

**Figure 1 ijerph-18-06144-f001:**
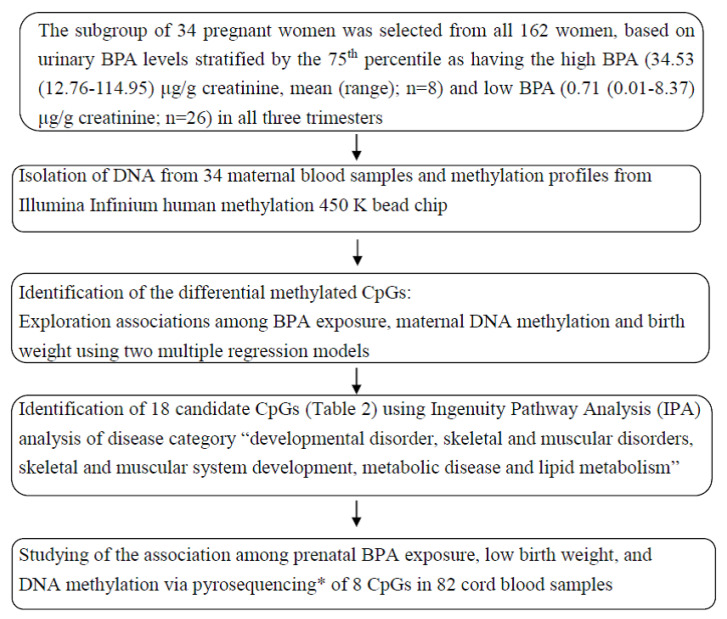
Flow chart in this study * bisulfite treatment and quantitative PCR (q-PCR).

**Table 1 ijerph-18-06144-t001:** Population characteristics (Mean (range)).

Variables	Total Subjects (*n* = 162)	Subjects with Illumina DNA Methylation Levels (*n* = 34)	*p*-Value
Maternal age (year)	32.2 (20–43)	32.6 (22.7–38.1)	0.90
Pre-pregnancy BMI (Kg/m^2^)	21.1 (15.4–34.1)	20.9 (17.3–28.2)	0.57
Maternal weight gain (Kg)	12.4 (8.6–16.2)	12.6 (8.0–19.0)	0.93
Birth weight (g)	3121 (2400–3880)	2928 (2520–3880)	0.15
BPA levels (μg/g creatinine)			
1st trimester	2.46 (0.01–46.5)	2.64 (0.01–24.1)	0.74
2nd trimester	4.53 (0.01–85.6)	6.14 (0.01–32.4)	0.30
3rd trimester	11.31 (0.02–153.0)	17.23 (0.03–114.9)	0.61
BPA stratified by the 75th percentile * Low exposure group			
1st trimester (<1.4 μg/g creatinine)	0.10 (0.01–1.3)	0.71 (0.01–8.4)	0.37
2nd trimester (<3.3 μg/g creatinine)	0.51 (0.01–3.3)		0.48
3rd trimester (<5.0 μg/g creatinine	0.31 (0.01–4.3)		0.86
High exposure group			
1st trimester (>1.4 μg/g creatinine)	9.72 (1.4–46.5)	34.53 (12.8–114.9)	NA
2nd trimester (>3.3 μg/g creatinine)	18.07 (3.7–85.6)		NA
3rd trimester (>5.0 μg/g creatinine)	41.6 (5.0–153.0)		0.38

Sample sizes in subjects with Illumina DNA methylation were 10, 10, and 14 in the first, second, and third trimesters, respectively. * In the subgroup (*n* = 34), prenatal BPA exposure groups were categorized into low (< 9.5 μg/g creatinine) and high (> 9.5 μg/g creatinine) exposure group based on the 75th percentile. Among them, 8 subjects were in the high BPA exposure group; 1, 2, and 5 were in the first, second, and three trimesters. In total subjects, prenatal BPA exposure groups were categorized into low and high exposure group based on the 75th percentile in the 1st trimester (1.4 μg/g creatinine), 2nd trimester (3.3 μg/g creatinine), and 3rd trimester (5.0 μg/g creatinine). NA: not available.

**Table 2 ijerph-18-06144-t002:** 18 CpG sites related with developmental disorder, skeletal and muscular disorders, skeletal and muscular system development, metabolic disease, and lipid metabolism.

Probe ID	CHR	Arm	Gene	Feature	CpG Islands	No. of CpG	Gene Name
cg18854735	1	p	GNB1	TSS1500	island	6	guanine nucleotide binding protein (G protein), beta polypeptide 1
cg24529814	1	p	PRDM16	Body	island	4	PR domain containing 16
cg23231974	1	p	PLOD1	1stExon	island	7	procollagen-lysine, 2-oxoglutarate 5-dioxygenase 1
cg10635194	1	p	KCND3	TSS200	island	6	potassium channel, voltage gated Shal related subfamily D, member 3
cg00251716	1	q	SDCCAG8	Body	open sea	1	serologically defined colon cancer antigen 8
cg03706175	2	p	EPCAM	Body	shore	2	epithelial cell adhesion molecule
cg27640254	2	q	DHRS9	TSS1500	open sea	2	dehydrogenase/reductase (SDR family) member 9
cg11543686	2	q	SLC19A3	Body	open sea	2	solute carrier family 19 (thiamine transporter), member 3
cg05524038	5	q	CSF1R	TSS1500	open sea	2	colony stimulating factor 1 receptor
cg18031134	6	p	HLA-G	Body	island	5	major histocompatibility complex, class I, G
cg19768311	6	p	DST	3’UTR	open sea	1	dystonin
cg01502353	6	p	DST	Body	open sea	1	dystonin
cg23244463	8	q	KCNB2	Body	open sea	4	potassium channel, voltage gated Shal related subfamily D, member 2
cg07349217	8	q	TG	TSS1500	open sea	1	thyroglobulin
cg19427642	10	q	KCNMA1	Body	open sea	1	potassium channel, calcium activated large conductance subfamily M alpha, member 1
cg27420224	20	q	HNF4A	TSS200	open sea	4	hepatocyte nuclear factor 4, alpha
cg00636769	20	q	GNASAS	Body	open sea	1	GNAS antisense RNA 1
cg17527673	22	q	SCARF2	Body	island	4	scavenger receptor class F, member 2

**Table 3 ijerph-18-06144-t003:** DNA methylation percentage of 8 CpGs in cord blood samples (*n* = 82) validated by q-PCR.

Probe ID	Gene	No. of CpG	Average DNA Methylation (%)
cg01502353	DST	1	88
cg05524038	CSF1R	2	CpG1^st^ (90) CpG2^nd^ (93)
			1^st^ and 2^nd^ CpG (91)
cg07349217	TG	1	96
cg19427642	KCNMA1	1	90
cg19768311	DST	1	91
cg23244463	KCNB2	4	CpG1^st^ (93) CpG2^nd^ (100)
			CpG3^rd^ (98) CpG4^th^ (100)
			Average of the above CG site (97)
cg27420224	HNF4A	4	CpG1^st^ (86) CpG2^nd^ (71)
			CpG3^rd^ (58) CpG4^th^ (77)
			Average of the above CpG (79)
cg27640254	DHRS9	2	CpG1^st^ (94) CpG2^nd^ (82)
			1^st^ and 2^nd^ CpG (88)

DST: dystonin; CSF1R: colony stimulating factor 1 receptor; TG: thyroglobulin; KCNMA1: potassium channel, calcium activated large conductance subfamily M alpha, member 1; KCNB2: potassium channel, voltage gated Shal related subfamily D, member 2; HNF4A: hepatocyte nuclear factor 4, alpha; DHRS9: dehydrogenase/reductase (SDR family) member 9.

**Table 4 ijerph-18-06144-t004:** Adjusted odds ratios (95% CI) between cord DNA methylation levels (cutoff 50^th^ percentile) at 8 CpG sites and low birth weight (<2500 g) calculated in an adjusted logistic regression model.

Genes	DNA Methylation Levels at CpG Sites (Cutoff by Median)	Odds Ratio	95% CI	*p*-Value
DST	cg01502353 (methylation >88% vs. <88%)	1.06	0.99–1.14	0.09
CSF1R	cg05524038 CpG^2nd^ (methylation >93% vs. <93%)	1.03	0.95–1.11	0.50
	cg05524038 1^st^ and 2^nd^ CpG sites (methylation >91% vs. <91%)	0.97	0.89–1.04	0.37
TG	cg07349217 (methylation >96% vs. <96%)	1.01	0.94–1.09	0.74
DST	cg19768311 (methylation >91% vs. <91%)	0.96	0.89–1.03	0.23
KCNB2	cg23244463 CpG^3rd^ (methylation >98% vs. <98%)	1.00	0.94–1.09	0.82
	cg23244463 1 to 4 CpGs (methylation >96.8% vs. <96.8%)	1.00	0.94–1.08	0.86
HNF4A	cg27420224 CpG^1st^ (methylation>86% vs. <86%)	0.97	0.90–1.04	0.40
	cg27420224 1 to 4 CpGs (methylation >79% vs. <79%)	1.01	0.94–1.09	0.74
DHRS9	cg27640254 CpG^2nd^ (methylation >82.5% vs. <82.5%)	1.10	1.00–1.21	0.047
	cg27640254 1^st^ and 2^nd^ CpGs (methylation >88.5% vs. <88.5%)	0.96	0.88–1.03	0.26

cg19427642 site: not available due to small sample sizes in DNA methylation level. Model adjusted for maternal age, gestational age, pre-pregnancy BMI, birth sex, and parity.

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
