# Peer review of "Prenatal Bisphenol a Exposure, DNA Methylation, and Low Birth Weight: A Pilot Study in Taiwan"

_ijerph, 2021, doi:10.3390/ijerph18116144_

Round 1

Reviewer 1 Report

The present research article describes the exposures of Bisphenol A causes increased risk of abnormal birth outcomes like low birth weight, and these could be mediated through the DNA methylation. However, the effects of Bisphenol A exposure on fetus development and DNA methylation are well studied earlier. There are many  published reports available in the literature which describes these abnormal births when exposures to BPA. Some of published papers related to this research are below:

Ye et al,  FASEB J. 2019 33(2): 2732–2742.

Choi et al, Environment International.2020, 143, 105929

Susiarjo et al, POLS Genetics, 2013, 9, e1003401.

Yang et al., Int J Environ Res Public Health. 2020 Jan; 17(1): 298.

Dai et al, Environmental Toxicology and Pharmacology,2016,  48, 265-271.

Müller et al, Scientific Reports 2018, 8, 9196.

Guida et al, Mutat Res. 2015, 774:33-9.

I did not see any major advantages in the present over earlier reports. I would suggest authors to clearly discuss about the published work and merits of present work.

Minor: Page 4: Please provide the gradient program used for the LC-MS analysis.

Author Response

  1. Response:Thank you for the comments regarding our manuscript. We have reviewed the published papers mentioned from the reviewer as follows. Four animal studies have reported that BPA exposure can provoke intrauterine grow restriction (Müller et al, 2018), cause preeclampsia-like features in pregnant mice involved in reprogramming of DNA methylation of WNT2 (Wnt family member-2; Ye et al., 2019), cause toxicity in spermatocytes and DNA methylation may exert an important role in BPA-triggered male reproductive toxicity (Dai et al, 2016), and alter the methylation levels of differentially methylated regions (DMRs) including the Snrpn imprinting control region (ICR) and Igf2 DMR1(Susiarjo et al, 2013). In addition, three human and epidemiological studies have shown that BPA can cause abnormal development of the central and peripheral nervous system in fetus (Guida et al, 2015) and BPA exposure on children asthma might through the alteration of MAPK1 methylation (Yang et al., 2020; also cited in this study in reference 42). Another epigenome-wide association study in children, in the high BPA compared to low BPA exposure groups, categorized by the 80th percentile of maternal BPA levels (2.7μg/g creatinine), is associated with an increase in DNA methylation of insulin-like growth factor 2 receptor (IGF2R). Notably, the increased methylation of IGF2R at age 2 was associated with BMI during age 2-8 years only in girls (Choi et al, 2020, [14]) (lines 68-72). Although previous experimental and human epidemiologic studies reported the effects of BPA exposure on fetal development and DNA methylation, there were different approaches of methodology or interested endpoint of health from our present study. Data on the role of maternal BPA exposure in relation to maternal/fetal DNA methylation in the blood and epigenetic changes linked to newborn birth outcomes are very limited. Thus, we were motivated to explore the epigenetic effects of BPA exposure on DNA methylation and elucidate the effects of methylation on low birth weight. This study has several strengths. First, we used multiple urine sampling points during pregnancy to explore the exposure to BPA and a critical window of susceptibility for fetal development. Second, we applied a novel approach: using an Illumina assay on multiple blood samples to identify candidate CpG methylation sites, followed by IPA and confirmation of the candidate CpG methylation sites by pyrosequencing. This step-wise approach is cost-effective and can help determine the most promising CpG sites responsible for the association between BPA exposure and low birth weight (line 345-351).

2. Response: Chromatography was performed using binary mixtures of 0.1% ammonia in water (mobile phase A) and 0.1% ammonia in methanol. The elution gradient conditions were as follows: 80% A for 0.5 min, 80%−1% A for 0.5 min, 1% A held for 1 min, 1%−80% A for 1 min, and re-equilibration at 80% A for 3 min (lines 122-124).

Reviewer 2 Report

Authors have achieved improvement of the manuscript by adding several analyses/detailed information/discussion. Even if still methods aspects are lacking, this is adequately discussed as raised as limitations. The manuscript could now be considered for publication after minor edits.

L149: Why probes with detected p-value < 0.01 were excluded?

L242 and 245: “…the first, second, and third trimesters…”

L262: “were selected for methylation validation by…” sounds like chosen manually, I would include that it was due to primer quality issues as mentioned for example in the abstract

Ll269ff: “After controlling for covariates, significant inverse associations were found between prenatal BPA exposure in the second trimester and low birth weight.” Due to the wording “inverse association” this sounds like the higher the maternal BPA the lower the prevalence of LBW?  Authors might also want to refer to specific data/table here to clarify.

L320: The level of BPA in amniotic fluid is lower than that in maternal urine, suggesting…

L430: “…lymphocytes, monocytes, granulocytes, and neutrophils.

Author Response

 L149: Why probes with detected p-value < 0.01 were excluded?

    Response: We apologized for the typo in the description. One of the internal filtering strategies of ChAMP is to perform filtering using detection p-values. By default, this is set to exclude probes that have a detection p-value > 0.01. We did not change this default setting and therefore the correct description should be: “Probes meeting any of the following conditions were removed: (1) internal controls, (2) detected p-value < > 0.01 (line 141)”.

L242 and 245: “…the first, second, and third trimesters…”

    Response: We have revised “the first, second, and three third” trimesters (line 229).

L262: “were selected for methylation validation by…” sounds like chosen manually, I would include that it was due to primer quality issues as mentioned for example in the abstract

   Response: We have revised “8 CpG sites met the primer design score requirements for methylation validation by bisulfite pyrosequencing were selected” (lines 247-248).

Ll269ff: “After controlling for covariates, significant inverse associations were found between prenatal BPA exposure in the second trimester and low birth weight.” Due to the wording “inverse association” this sounds like the higher the maternal BPA the lower the prevalence of LBW?  Authors might also want to refer to specific data/table here to clarify.

Response: After controlling for covariates, each natural-log increase in urinary BPA level (μg/g creatinine) in the second trimester was associated with a 0.02 g decrease in low birth weight (p <0.05) (lines 253-254).

L320: The level of BPA in amniotic fluid is lower than that in maternal urine, suggesting…

Response: We have revised “The level of BPA in amniotic fluid is lower than that in maternal urine” (line 291).

L430: “…lymphocytes, monocytes, granulocytes, and neutrophils.

Response: We have revised “neutrophils” (line 369).

This manuscript is a resubmission of an earlier submission. The following is a list of the peer review reports and author responses from that submission.

Round 1

Reviewer 1 Report

This research paper describes the prenatal Bisphenol exposure, and risk factors. The manuscript is well written and discussion well.

I have few minor comments:

Section 2.2: BPA and creatinine measurement:  Please provide the  UPLC-TOF operating, and  chromatographic conditions and extraction procedures.

Similarly, please add brief details about the creatinine analysis like reaction conditions and spectrophotometer conditions etc.

Minor: Please correct the section number in line 81, 96 …in page 3 & 4.

Minor: Conclusion can be improved.

Author Response

Section 2.2: BPA and creatinine measurement:  Please provide the  UPLC-TOF operating, and  chromatographic conditions and extraction procedures.

Response 1: We thank the reviewer for this constructive comment. We have added the analysis of BPA and creatinine measurement. Please see lines 110-125.

“Sample pretreatment and quantification of BPA followed previous procedures [10]. In brief, urine samples (two milliliters) were spiked with 50 mL of 13C12-BPA (2000 ng/mL), acidified to pH 5.5, deconjugated by β-glucuronidase (Sigma-Aldrich, USA), incubated at 37 oC for 15 h, and then acidified to pH 3 with 1 M hydrochloric acid. Next, the samples were extracted with PH solid-phase extraction cartridge and were eluted with 1 mL methanol. BPA was analyzed using ultra-performance liquid chromatography coupled with time-of-flight mass spectrometry (Waters, MA, USA). Separation was achieved using a BEH C18 column (1.7 μm, 2.1 × 10 mm, Waters, USA) with a flow rate of 0.35 mL/min. Chromatography was performed using binary mixtures of 0.1% ammonia in methanol and 0.1% ammonia in water. The TOF/MS was operated by monitoring ion mass transitions as follows: m/z 227.12→133.07 for BPA and m/z 239.18→139.09 for 13C12-BPA. The average recovery for BPA was 93-100% at the 20-100 ng/mL level. The limit of detection (LOD) for BPA in urine was 0.16 ng/mL. Creatinine was analyzed based on modifications of the Jaffe reaction made by Hinegard and Tiderstrom [2]. Briefly, 0.1 mL of each urine sample was added to 3 mL 3.3 mM picric acid and mixed with 0.17 M sodium hydroxide and 26 mM sodium tetraborate. This mixture was then incubated at 37°C for 15 min and measured with a spectrophotometer at a wavelength of 510 nm.”

Minor: Please correct the section number in line 81, 96 …in page 3 & 4.

Response 2: We thank the reviewer for this constructive comment. We have corrected it in the manuscript.

Minor: Conclusion can be improved.

 Response 3: We thank the reviewer for this constructive comment. We have revised the conclusion. Please see lines 360-361.

Reviewer 2 Report

Manuscript ID: ijerph-940634

Title: Prenatal Bisphenol A Exposure, DNA methylation, and Low Birth Weight: A Pilot Study in Taiwan

This study aimed to investigate the effects of maternal BPA exposure (analysed in the urine) on maternal/fetal DNA methylation levels and the associations with low birth weight. The study is presenting an interesting topic, pointing to an interesting potential link to already shown prenatal BPA driven overweight/obesity risk in infancy. However, there are some major issues that need to be addressed. Authors need to proof that shown maternal methylation changes indeed have a further physiological impact.

Methylation changes are analyzed genome wide via 450K assay in a quite small cohort (n=34) and no methylation validation is provided in maternal samples (e.g. pyrosequencing) nor is a functional relevance proven by mRNA expression (e.g. PCR) or protein level in either maternal or cord blood samples. Authors were aiming to validate BPA driven methylation changes via pyrosequencing in cord blood, however, cord blood data were not well correlated with maternal 450K data.

Further comments:

Introduction:

  • If literature is checked for ‘prenatal BPA - methylation - cohort` there are indeed several studies supporting the hypotheses that BPA is modifying the epigenetic pattern of weight associated genes. This part could be outlined in more detail to support the authors hypothesis since a BPA driven low birth weight might be resulting in overweight/obesity in childhood.

Methods:

  • How were the 34 samples selected? If according to BPA exposure, why only 6 high samples were chosen?
  • Adjustments for cell composition of needs to be included in the 450K data to account that methylation changes are not due to alterations in the (white) blood cell composition
  • Ll1134: to what is this p-value referring to?
  • Ll115: why Y chromosome in maternal samples?
  • Ll 134: indicate the used FDR
  • In general: indicate clearly which methylation changes are described/reported: maternal/cord blood (entire text, tables etc)
  • Please indicate better if you are talking about single CPGs or CPG islands (e.g. ll 156; and describe the definition for CPG islands)

Results/Discussion:

  • It might be discussed why the majority of the 8 identified CPGs/genes are generally quite hypermethylated (80% and more).
  • Ll170: no significant difference… please provide data/p-values

Table 1:

  • Needs some more formatting (e.g. include “trimester” after 1st/2nd/3rd, include “exposure group” after low/high
  • Is it always the geometric mean shown in the most right column?
  • Also include BPA levels for the 162 total cohort (high/low)
  • Low/high group is 26 and 8 here? Elsewhere it is described as 28 vs. 6 samples
  • Please provide a p-value for the comparison of the 34 vs 162 samples
  • BPA levels are increasing with increasing trimester, this should be discussed
  • Indicate, how many subjects with high BPA levels (>75% percentile) are in the 3rd trimester (maybe the seen methylation changes are just because there is a very high overlap with 3rd trimester samples. Therefore, it is even more of importance to adjust for blood cell composition since it is known immune cells change tremendously throughout pregnancy)
  •  

Table 2:

  • could/should? be in the supplements
  • footnote: confounders seem to be listed twice – aren’t they the same for both models?

Discussion:

  • Ll220-227 should be in the results part: This is an important point to provide evidence for the plausibility of the data. However, authors need – at least - provide method validation of the 450K data e.g. via pyrosequencing in maternal samples which should have a nice correlation. Further, these data could then be correlated with cord blood sample pyroseq data.

Author Response

 Major comments:

Response 1: We thank the reviewer for this constructive comment. We used multiple urine sampling points during pregnancy to explore the exposure to BPA and a critical window of susceptibility for fetal development and applied a novel approach using an Illumina assay on blood samples to identify candidate CpG methylation sites, followed by IPA and confirmation of the candidate CpG methylation sites by pyrosequencing in cord blood. This step-wise approach is cost-effective and can help determine the most promising CpG sites responsible for the association between BPA exposure and low birth weight. We also acknowledged the limitations that no available data on mRNA expression or cell type distribution of the blood samples.

In the present study, we observed nonsignificant associations between maternal and cord DNA methylation, except for a significant correlation between maternal and cord blood DNA methylation in DST gene using Illumina assay (rsp=0.73, p=0.01; Supplemental Figure S1). In addition, nonsignificant associations were observed between cord DNA methylation measure using Illumina average beta values (y axis) and cord pyrosequencing (x axis) for 8 CpG sites: (a) cg01502353_DST (n=6, r= 0.32; p=0.54); (b) cg05524038_CSF1R (n=6, r= –0.09 ; p=0.87);(c) cg07349217_TG (n=6, r=–0.20; p=0.70);(d)cg19427642_KCNMA1 (n=5, r=0.30; p=0.62);(e) cg19768311_DST (n=6, r=–0.35; p=0.49);(f) cg23244463_KCNB2 (n=6, r=–0.17; p=0.74); (g) cg27420224_HNF4A (n=6, r=0.49; p=0.33); (h) cg27640254_DHRS9 (n=5, r=0.05; p=0.93).

The figures were supplemented in the attachment.

Further comments:

Introduction:

Response 2: We thank the reviewer for this constructive comment. We have added references. Please see lines 61-68.

“An epigenetic epidemiological study showed that prenatal BPA exposure causes hypomethylation in the obesity-associated mesoderm-specific-transcript (MEST) gene promoter in cord blood as well as increased expression of MEST, which contributes to increase in the body mass index (BMI) Z-score in children until the age of 6 years [13]. Another epigenome-wide association study in children, in the high BPA compared to low BPA exposure groups, categorized by the 80th percentile of maternal BPA levels (2.7 μg/g creatinine), is associated with an increase in DNA methylation of insulin-like growth factor 2 receptor (IGF2R). Notably, the increased methylation of IGF2R at age 2 was associated with BMI during age 2-8 years only in girls [14]”.

Methods:

How were the 34 samples selected? If according to BPA exposure, why only 6 high samples were chosen?

Response3: Thirty-four samples were selected based on urinary BPA levels in all three trimesters stratified by the 75th percentile (9.5 μg/g creatinine), as high BPA exposure group [mean (range) levels with 34.53 (12.8-114.9) μg/g creatinine; n=8] and low BPA exposure group [mean (range) levels with 0.71 (0.01-8.4) μg/g creatinine; n=26]. We have corrected 8 samples in BPA high exposure group throughout the manuscript.

Adjustments for cell composition of needs to be included in the 450K data to account that methylation changes are not due to alterations in the (white) blood cell composition

Response 4: We thank the reviewer for this constructive comment. We recognized that cord blood differed from adult peripheral blood in physiological states, levels of immune cell maturity, and cell type compositions. The study subjects we recruited were healthy and cell counts were considered in a normal range. The benchmark of estimated cell type proportions was equal to matched cell counts. Since different types of white blood cells might be responsible for the DNA methylation changes (Gervin et al. 2016), the confounding effect of cell proportion need to be adjusted. However, we did not perform differential blood cell counts and no data on whether specific white blood cells were highly methylated and established the linkage with birth weight. We have acknowledged this limitation. Please see lines 351-357.

Ll134: to what is this p-value referring to? indicate the used FDR?

Response 5: For multiple comparisons, p value was adjusted by the false discovery rate (FDR), which was a FDR-corrected p value, according to the Benjamini and Hochberg (1995) method [5] (lines 156-158).

Ll115: why Y chromosome in maternal samples?

Response 6: We thank the reviewer for this comment. We have deleted Y chromosome in maternal samples (line138).

In general: indicate clearly which methylation changes are described/reported: maternal/cord blood (entire text, tables etc)

Response 7: We thank the reviewer for this constructive comment. We have revised this throughout the manuscript.

Please indicate better if you are talking about single CPGs or CPG islands (e.g. ll 156; and describe the definition for CPG islands)

Response 8: We thank the reviewer for this comment. We have checked and revised as single CPGs (not CPG islands) (line 180) and throughout the manuscript.

Results/Discussion:

It might be discussed why the majority of the 8 identified CPGs/genes are generally quite hypermethylated (80% and more).

Response 9: As the nature of growth-related genes during fetal development, the DNA methylation level in cord blood is hypermethylated, which is consistent with those reported in the literature [39,40].

Ll170: no significant difference… please provide data/p-values

 Response 10: We thank the reviewer for this comment. We have provided p-values in Table1.

Table 1:

Needs some more formatting (e.g. include “trimester” after 1st/2nd/3rd, include “exposure group” after low/high

Is it always the geometric mean shown in the most right column?

Also include BPA levels for the 162 total cohort (high/low)

Please provide a p-value for the comparison of the 34 vs 162 samples

Response: We have revised in Table 1.

BPA levels are increasing with increasing trimester, this should be discussed

Response: In this study, the increasing trend of BPA throughout the three trimesters may be attributed to the changes in dietary behavior (e.g., the intake of meat, vegetables, and fruit) and lifestyle (e.g., frequency of using plastic products and detergent). Please see lines 279-281.

Indicate, how many subjects with high BPA levels (>75% percentile) are in the 3rd trimester (maybe the seen methylation changes are just because there is a very high overlap with 3rd trimester samples. Therefore, it is even more of importance to adjust for blood cell composition since it is known immune cells change tremendously throughout pregnancy)

Response: We have indicated that 62% (5/8) subjects with high BPA levels are in the 3rd trimester in Table 1 and acknowledged as limitation of this study. Please see lines 351-357.

Low/high group is 26 and 8 here? Elsewhere it is described as 28 vs. 6 samples

Response: Low/high group is 26 and 8 and we have revised throughout the manuscript.

Table 2:

could/should? be in the supplements

footnote: confounders seem to be listed twice – aren’t they the same for both models?

 Response: We thank the reviewer for this constructive comment. We have revised Table S2 and footnote in supplementary Table S2.

Discussion:

Ll220-227 should be in the results part: This is an important point to provide evidence for the plausibility of the data. However, authors need – at least - provide method validation of the 450K data e.g. via pyrosequencing in maternal samples which should have a nice correlation. Further, these data could then be correlated with cord blood sample pyroseq data.

Response: We thank the reviewer for this constructive comment. We have placed Ll220-227 in the results part (lines 255-263). Please see response 1 for the method validation.

Reviewer 3 Report

The authors Yu-Fang Huang et. al.  in their manuscript tilted “Prenatal Bisphenol A Exposure, DNA methylation, 2 and Low Birth Weight: A Pilot Study in Taiwan” have presented their data on epigenomic profiling of BPA exposed mothers, and the outcome of BPA exposure on the new born. The authors have employed DNA methylation of whole blood DNA, and found that BPA exposure was associated with low birth weight. Additionally, they have found that dehydrogenase/reductase member 9 (DHRS9) gene was hypermethylated in high BPA exposed mothers. Thus, the authors conclude that BPA and DHRS9 contribute to low birth weight.

The manuscript is well written and the results are presented in a lucid manner. However, an explanation to the following concerns would make the manuscript more interesting.-

1 How much BPA was there in blood or it is correlative of the amount that is found in urine.

  1. Is there a possibility that the amount of BPA in amniotic fluid is lesser/higher than that urine?

  1. The authors did not find any change in the cord blood DNA methylation levels. Is it may be because the concentration of BPA could be considerably low as compared to urine?

Author Response

Response: We thank the reviewer for this constructive comment. We have added and discussed the BPA levels in tissue distribution. Please see lines 298-304. “One study reported the order of BPA levels in tissue distribution (e.g., maternal serum and urine, neonatal urine and cord serum) are urine in mother and neonates > cord serum> maternal serum [8]. In addition, BPA in cord serum, significantly associated with maternal serum and urine. Amniotic fluid is mainly derived from fetal urine and the mass exchange with umbilical cord during the 3rd trimester. The level of BPA in amniotic fluid is lower than that maternal urine, suggesting that the metabolism of BPA was low in the fetus [36, 37]”.

Round 2

Reviewer 2 Report

Unfortunately, authors were not able to address the major concerns regarding

  • A technical validation of methylation patterns in mothers (that would be necessary for such a small n-number of 34) nor showing a functional relevance of the methylation changes in either mother/and or children.
  • Further, authors didn't provide adjustments for cell composition which is of need to be included in the 450K data to account that methylation changes are not due to alterations in the blood cell composition, especially if samples for 450K analyses are used from different stages throughout pregnancy. We acknowledge that there are no cell counts available, but there are other methods based on the 450K data that could be used to overcome this point by using a reference data set consisting of cell type specific DNA methylation signatures (see for example Gervin et al. 2016 for cord blood data etc.)

So from my side, data are not free from doubt to be published in the present form.